# No Association of Angiotensin-Converting Enzyme Insertion/Deletion (ACE I/D) Gene Polymorphism in the Susceptibility to Diabetic Retinopathy in Type 2 Diabetes Mellitus Patients: An Updated Meta-Analysis

**DOI:** 10.3390/jpm13091308

**Published:** 2023-08-26

**Authors:** Aline Ruilowa de Pinho Coelho, Luciana Carvalho Silveira, Kamilla de Faria Santos, Rodrigo da Silva Santos, Angela Adamski da Silva Reis

**Affiliations:** 1Laboratory of Molecular Pathology, Biological Sciences Institute, Federal University of Goiás, Goiânia 74690-090, Brazil; 2Department of Biochemistry and Molecular Biology, Biological Sciences Institute, Federal University of Goiás, Goiânia 74690-090, Brazil

**Keywords:** *ACE* gene, diabetic retinopathy, genetic polymorphism, type 2 diabetes mellitus

## Abstract

Diabetic retinopathy (DR) is a complex and multifactorial pathology encompassing environmental, metabolic, and polygenic influences. Among the genes possibly involved in the development and progression of DR, the *Angiotensin I-converting enzyme* (*ACE*) gene stands out, which presents an insertion (I) or deletion (D) polymorphism of a 287 bp Alu repetitive sequence in intron 16. Thus, this study aimed to perform a systematic review with meta-analysis to elucidate the relationship between the *ACE* gene (I/D) polymorphism (rs1799752) and the development and progression of DR in type 2 diabetic patients. PubMed/MEDLINE, Embase, Web of Science, and Scopus databases were systematically searched to retrieve articles that investigated the association between *ACE* gene (I/D) polymorphism in DR patients. Sixteen articles were included in the systematic review. The results describe no significant association between the polymorphism and DR risk (OR = 1.12; CI = 0.96–1.31; and *p* = 0.1359) for genotypic analysis by the dominant model (II vs. ID+DD). Moreover, we also observed no significant association between the D allele on the allele frequency analysis (I vs. D) and the DR risk (OR = 1.10; CI = 0.98–1.23; and *p* = 0.1182). Forest plot analysis revealed that the discrepancy between previous studies most likely arose from variations in their sample sizes. In conclusion, I/D polymorphism appears to be not involved in the susceptibility to and progression of the DR in type 2 diabetic patients.

## 1. Introduction

Diabetic retinopathy (DR) is a common microvascular complication of diabetes mellitus (DM), classified as one of the most recurrent causes of blindness and visual impairmentworldwide. However, its etiology has not been fully elucidated [1]. Furthermore, DR is usually asymptomatic in the early stages, highlighting the need for regular ophthalmologic exams that confirm the diagnosis and enable the subsequent management of the disease [2]. There are two categories of DR: non-proliferative diabetic retinopathy (NPDR) and proliferative diabetic retinopathy (PDR) [3].

The increased permeability of capillary pericytes and vascular endothelial damage determine the appearance of microaneurysms and dot intraretinal hemorrhages, which are the main characteristics of NPDR. With disease progression, vasoconstriction and capillary occlusions form tortuous capillaries and retinal ischemia, as well as cotton wool spots that can also be observed at this stage. In the final stage of the disease (PDR), severe hypoxia causes neovascularization, vitreous hemorrhage, and retinal detachment [2,3].

DR is a complex and multifactorial pathology encompassing environmental, metabolic, and polygenic influences. Although the known risk factors are diabetes duration and glycemic control, not all patients with diabetes develop DR. Therefore, genetic variation may explain the heterogeneity in DR progression [4]. Among the genes possibly involved in the development and progression of DR, the *Angiotensin I-converting enzyme* (*ACE*) gene stands out, responsible for producing ACE that converts Angiotensin I into Angiotensin II, an important vasoconstrictor that induces the expression of Vascular Endothelial Growth Factor (VEGF), stimulating angiogenesis (increased intraocular angiogenesis that leads to PDR and diabetic macular edema) [5].

The *ACE* gene is located at 17q23.3 and has 26 exons and 25 introns. More than 160 polymorphisms were identified in this gene [6]. The most studied polymorphism of the *ACE* gene is characterized by the insertion or deletion (I/D) of a 287 bp Alu repetitive sequence in intron 16 (rs1799752). This polymorphism is associated with increased levels of circulating ACE and a higher activity of this enzyme [7]; the D allele was related to increased ACE activity compared to the presence of the I allele [8,9].

On the other hand, this polymorphism has been implicated in the pathogenesis of DM and its complications. Moreover, this polymorphism has been increasingly highlighted, mainly due to its relationship with the severity of COVID-19 in diabetic patients [10]. Thus, we aimed to perform a systematic review with meta-analysis to elucidate the relationship between the *ACE* gene (I/D) polymorphism (rs1799752) and the development and progression of DR in type 2 diabetes mellitus (T2DM) patients.

## 2. Materials and Methods

### 2.1. Registration and Search Strategy

Based on the guide question “What is the association between the I/D polymorphism of the *Angiotensin I Converting Enzyme* gene and the development and progression of diabetic retinopathy in patients with type 2 diabetes mellitus?”, a systematic review was conducted to determine the association between the *ACE* gene (I/D) polymorphism (rs1799752) and DR in T2DM patients. Publications from 1992 to 2022 were selected, comprising the last 30 years of analysis on the subject.

To avoid duplication, this study had the protocol registered in the International Prospective Register of Systematic Reviews (PROSPERO) on 12 December 2020 (number CRD42020215932) (Appendix A). For a better description of the data, we followed the Preferred Report Items for Systematic Reviews and Meta-analyses (PRISMA) guidelines [11].

The literature search occurred between June and December 2022 in the PubMed/MEDLINE, Embase, Web of Science, and Scopus databases. The last access to databases occurred on 13 December 2022. Combined terms for “Polymorphism I/D”, “ACE gene”, and “diabetic retinopathy” were used to elaborate the search strategy, being adapted for each database (Table 1). Filters restricting the last 30 years (1992–2022) were applied to all databases.

### 2.2. Inclusion and Exclusion Criteria

Inclusion: Observational studies that determined the genotypic distribution of the *ACE* gene (I/D) polymorphism (rs1799752) in DR patients, in the control (patients with T2DM and without DR) and/or healthy controls; in English; and published from 1992 to 2022.

Exclusion: Studies that did not address the *ACE* gene (I/D) polymorphism (rs1799752); another study model; and studies with patients with another type of DM.

### 2.3. Selection of Articles and Evaluation of Methodological Quality

The articles found in the search were included in the Rayyan platform [12]. Two independent reviewers performed the screening in two phases (I and II), in which the articles were evaluated according to predetermined inclusion and exclusion criteria. The first stage was performed by reading the title and abstract, and the second stage consisted of the whole reading of the selected articles in the first stage.

The risk of methodological bias was assessed using the Joanna Briggs Institute (JBI) critical assessment tools for each study design [13]. Only studies that responded positively to at least 70% of the questions were considered low risk of bias. The differences of opinion between the reviewers were discussed and resolved by consensus.

### 2.4. Data Extraction

Data extraction was performed by two reviewers independently in all studies included, and the following data were extracted: (1) authors and year of publication; (2) country of study; (3) study design; (4) sample size; (5) sexes of sample individuals; (6) average age of the sample; (7) clinical characteristics; and (8) genotypic and allele frequencies. Authors of studies with missing data were contacted.

### 2.5. Statistical Analysis

A meta-analysis was performed to evaluate the association of the *ACE* gene (I/D) polymorphism (rs1799752) with DR, using the calculation of the odds ratio (OR) and 95% confidence interval (CI), with the results presented in a forest plot. The ORs were calculated for the comparisons: (i) genotypic comparison with the dominant model (II vs. ID+DD); and (ii) allelic comparison (I vs. D). In addition, the heterogeneity between the studies was evaluated by the Higgins inconsistency test (I^2^).

The selection of the meta-analytical model was based on the result of the heterogeneity test between the studies. The fixed effect model (Mantel–Haenszel method) was applied when I^2^ < 25% (low heterogeneity), and the random effect model (DerSimonian-Laird method) was applied when I^2^ 25–75% (moderate heterogeneity). Values >75% were defined as high heterogeneity. If necessary, sensitivity analysis will be performed to evaluate the possible sources of heterogeneity among the studies.

In addition, the Hardy–Weinberg equilibrium (HWE) was calculated using Fisher’s exact test. In meta-analyses of genetic association studies, HWE deviations in controls have been linked to problems in the design and conduct of the studies, mainly due to population stratification, genotyping error, or selection bias.

Publication bias was assessed by Egger’s regression asymmetry test and Begg’s adjusted rank correlation test [14,15], visually displayed in funnel plots. A *p*-value <0.05 suggested a high probability of publication bias. All statistical analyses were performed in the RStudio^®^ software (version 4.1.0).

## 3. Results

### 3.1. Selection and Individual Results of Studies

We identified 2962 articles initially, and 1551 duplicates were removed. In stage I, reading titles and abstracts, 1392 articles did not meet the eligibility criteria. Thus, 19 studies were included in stage II. Nevertheless, three studies were excluded after the complete reading of the manuscript (Figure 1).

From 16 included studies, we had 1 cohort study, 4 cross-sectional studies, and 11 case–control studies. All selected articles were written in English. We identified two studies published in the years 1995, 2001, and 2003, respectively, and one article was published in each of the years 1997, 1998, 2000, 2005, 2006, 2008, 2010, 2014, 2018, and 2021. In this meta-analysis, 2423 (41.95%) individuals with DR (cases) and 3353 (58.05%) individuals without DR (controls) were genotyped. The characteristics of each study can be observed in Table 2. This section may be divided into subheadings. It should provide a concise and precise description of the experimental results, their interpretation, as well as the experimental conclusions that can be drawn. This table shows the compiled main data of the 16 eligible studies included in this systematic review.

In addition, the studies were homogeneous in terms of methodological quality assessment, individually reaching a minimum of 70% of positive responses, being considered with a low risk of bias (Table 3). For case–control studies, 9 parameters were evaluated for 8 cohorts and 11 cross-sectional parameters, respectively.

Table 4 describes the genotypic and allelic frequency of *ACE* gene (I/D) polymorphism found in the case and control groups of the included studies. Additionally, the distribution of genotypes in the control group of the included studies that agreed with the HWE (*p* = 0.06).

### 3.2. Meta-Analysis

Meta-analysis was performed with the 16 articles selected in the systematic review. Figure 2 describes a no significant association between the polymorphism and DR risk (OR = 1.12; CI = 0.96–1.31; and *p* = 0.1359) for genotypic analysis, by the dominant model (II vs. ID+DD). Moreover, we observed also no significant association between the D allele on the allele frequency analysis (I vs. D) and the DR risk (OR = 1.10; CI = 0.98–1.23; and *p* = 0.1182), Figure 3.

We observed that the larger the sample size the greater the weight of the study and the smaller the CI, indicating greater certainty in the results with a larger sample size (Figure 2 and Figure 3). Small samples may reflect observations due to chance, so the statistical indicators used in meta-analyses, such as CI and individual weight, confirm the degree of reliability of the results.

Our findings showed moderate heterogeneity in the Higgins inconsistency test, which did not interfere with the results obtained. However, we can observe the large variation in sample sizes of the included studies (Table 2), which may reflect the degrees of divergence found. In addition, the funnel plot estimated the publication bias for genotypic comparison (Figure 4) and allelic comparison (Figure 5), and no significant asymmetries were found to identify publication bias.

Egger’s regression asymmetry test and Begg’s adjusted rank correlation test were used to estimate the possible publication bias. Table 5 describes a significant difference in the genotypic comparison with Egger’s test (*p* = 0.01). However, no other indications of publication bias were found in the studies included in this meta-analysis. Results for a possible publication bias indicated that some studies may underestimate or overestimate the association found, in addition to reflecting a greater facility to publish positive results. However, these results must be interpreted in association with other variables, such as the sample size, study quality, and consistency of the results in independent studies.

## 4. Discussion

According to the International Diabetes Federation [31], DM affects more than 500 million adults worldwide, aged 20–79 years. The world ranking of countries with the highest number of diabetic adults in 2021 is China, India, and Pakistan [31]. Our systematic review highlighted studies conducted in Japan [16,19,20], United Kingdom [17], Spain [18], Turkey [21,25], Czech Republic [22], Slovenia [23], China [24,30], South Korea [26], Iran [27,28], India [29], and Jordan [5].

DR is one of the most devastating microvascular complications of DM and remains a leading cause of visual morbidity in both developed and developing countries [30]. Furthermore, all ethnicities are affected by DR. On the other hand, studies suggest that African/Afro-Caribbean, South Asian, Latin American, and indigenous tribal populations have relatively higher rates of DR [32]. In this systematic review, we only observed European and Asian studies, making it impossible to characterize and analyze specific genetic factors for each ethnicity, thus revealing the lack of studies analyzing the relationship between *ACE* gene (I/D) polymorphisms and the DR risk in different populations.

Hyperglycemia in diabetic patients triggers several biochemical processes, leading to inflammation, ischemia, and a pro-angiogenic state in the retina, with subsequent complications of vascular leakage, edema, neovascularization, and neurodegeneration [33,34]. Furthermore, this condition acts as an excitatory stimulus for the renin–Angiotensin–aldosterone system (RAAS), an endocrine hormonal cascade important for regulating blood pressure and hydroelectrolytic homeostasis [4,35]. In addition, the genes of this system play an important role in glucose metabolism [6].

The main components of this system are renin, angiotensinogen (AGT), Angiotensin I and II (Ang I and Ang II), Angiotensin-converting enzymes 1 and 2 (ACE and ACE2), and type 1 (AT1R) and type 2 receptors (AT2R). ACE hyperactivation increases the synthesis of the vasoconstrictor molecule Ang II [36]. Ang II alters glucose homeostasis by inhibiting insulin signal transduction, reducing glucose uptake, increasing insulin resistance, and destroying pancreatic β cells by inducing oxidative stress [37].

Polymorphic variants in genes acting in this system have been correlated with DM and its complications, and RAAS inhibition has reduced the incidence of diseases [6]. Hyperglycemia in DM patients increases Ang II levels, inducing oxidative stress and glomerular hyperfiltration, consequently leading to endothelial damage, thrombosis, inflammation, and vascular remodeling [38]. Abnormal ACE expression in the retina produces adverse effects on retinal blood flow and vascular structure [6]. Therefore, ACE inhibition has been associated with tissue protection of organs to improve ocular function in DM patients, consequently reducing the development and progression of DR [39,40].

Interestingly, *ACE* gene (I/D) polymorphism is responsible for approximately 50% of the phenotypic variations found in plasma ACE levels, and individuals with the I/I genotype have reduced levels of ACE, whereas I/D individuals have intermediate levels, and D/D individuals have increased levels of the enzyme [8]. Thus, several studies have sought the relationship between RAAS genes and the pathogenesis of DM and its complications, such as DR [41,42,43]. However, the results are still controversial.

Feghhi et al. [27] found a significant association in the I/I vs. D/D comparison (OR = 3516, CI = 1279–9665, and *p* = 0.015), whereas Nikzamir et al. [28] identified an association in the I/D vs. I/I comparison (OR = 1831, CI = 1074–3124, and *p* = 0.026) in Iranian patients with T2DM and DR. Furthermore, Matsumoto et al. [20] associated the D allele with the PDR (*p* = 0.041). Additionally, in Pakistanis, I/D polymorphism was significantly associated with NPDR but not with PDR [43].

Luo et al. [4] performed a meta-analysis evaluating the association between the *ACE* gene (I/D) polymorphism and the DR risk and found a significant association. Additionally, the authors also described a significant association between the polymorphism studied and the DR risk for all genetic models. However, this study included different types of DM and other complications, such as diabetic nephropathy (DN). Differently, our meta-analysis assessed the susceptibility to DR only in T2DM patients for I/D polymorphism. Previous studies have demonstrated the association of DN with the D allele in this polymorphism [44,45,46], demonstrating a possible selection bias and/or confusion in the meta-analysis performed by Luo et al. [4].

These authors (Luo et al. [4]) still highlighted that *ACE* gene (I/D) polymorphism appears to be involved in the susceptibility to DR in the Asian population [4]. In contrast, Qiao et al. [30] analyzed 1491 patients with T2DM and was not found an association between *ACE* gene (I/D) polymorphism and DR in the Chinese population, as well as the other studies included in this systematic review and the meta-analyses performed by [47,48,49]. Another meta-analysis conducted by Zhou et al. [50] also described that the I/D polymorphism was unrelated to the pathogenesis of T2DM in the Chinese population. Considering that most of the studies included in our meta-analysis are from the Asian population, *ACE* (I/D) polymorphism appears probably not to contribute to DR development, especially in Asian patients with T2DM.

In the context of the non-association between I/D polymorphism and DR, our findings reaffirmed that this polymorphism is not associated with DR susceptibility or with its progression in patients with T2DM. Accordingly, the Higgins inconsistency test did not indicate high heterogeneity between studies. Therefore, we can observe that the studies included in the meta-analysis showed a wide variation in sample size (Table 1), which may reflect the divergence of results found. Thus, when a small sample is used to represent the population of interest, the risk of observations due to chance is greater [51]. In this context, statistical indicators such as the confidence interval and the weight of each study in meta-analyses are used to better determine the degree of certainty of the results. In our data, we can observe that the larger the sample size the greater the weight of the study and the smaller the confidence interval (Figure 2 and Figure 3), indicating greater certainty in results with a larger sample size.

In addition, we found a significant difference in publication bias with Egger’s test for genotypic comparison; the other tests found no significant difference (Begger’s test and funnel plots) (Table 3, Figure 4 and Figure 5). This suggests that the results found in the individual studies may be overestimating or underestimating the true association. However, due to a statistical tool, the results should be interpreted in conjunction with other information, such as sample size, quality of studies, and consistency of results in independent studies. Thus, we highlight the importance of systematic reviews and meta-analyses such as this one, seeking to group, evaluate, and elucidate the true results and possible limitations of genetic association studies.

## 5. Conclusions

The limitations of this review could be described as (i) different sample sizes found: some of the studies analyzed presented relatively small case–control groups, and the minor allele frequency; (ii) possible publication bias found for genotypic analysis, indicating nonsignificant findings that may exist remaining unpublished, thereby artificially inflating the apparent magnitude of an effect; (iii) ethnicity: in meta-analyses of genetic association studies, ancestry and ethnicity are not accurately investigated; and (iv) the lack of studies evaluating other populations, which limited the possibility to further discussions. However, our findings provided consistent evidence for no associations between ACE gene (I/D) polymorphism and DR risk. Thus, our findings provide subsidies in the area of genetic epidemiology to help in the understanding of the genetic factors involved in the susceptibility to the development of DR.

This systematic review and meta-analysis revealed that *ACE* gene (I/D) polymorphism appears to be not involved in the susceptibility to DR or with its progression in patients with T2DM. Thus, no evident association was implied with overall DR risk in any genetic models. Additionally, meta-analyses, exploring sensitivity analyses, and possible publication biases are encouraged. The identification of genes and polymorphisms related to the development and progression of T2DM and its complications can help in the early diagnosis and development of effective treatments.

## Figures and Tables

**Figure 1 jpm-13-01308-f001:**
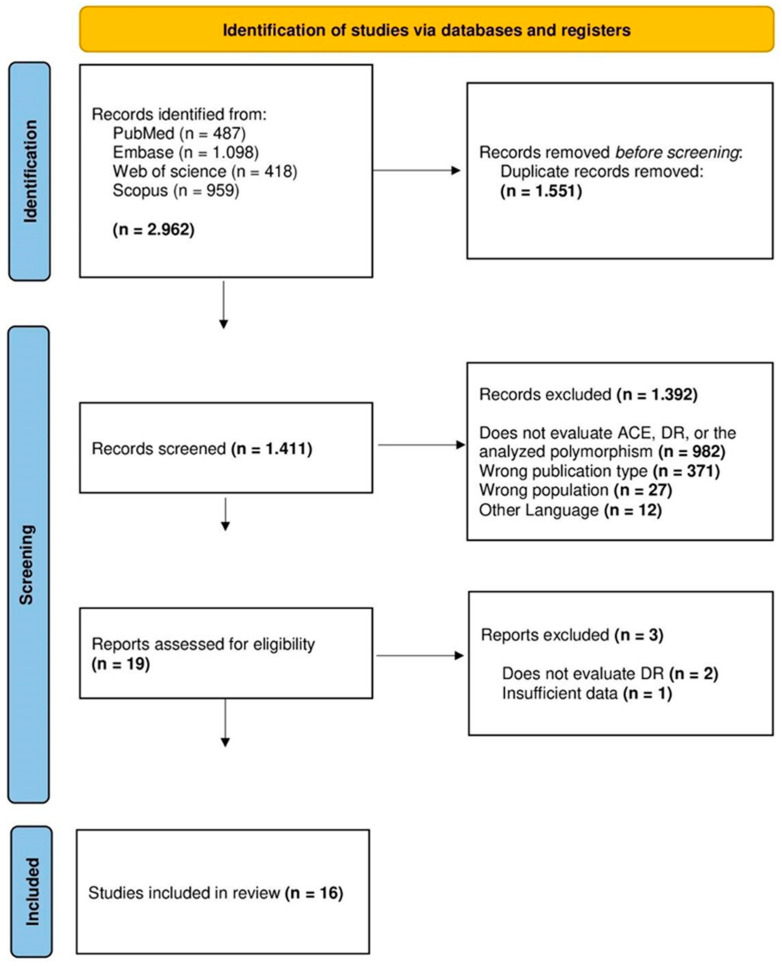
PRISMA flowchart detailing the selection process of the articles included and excluded in this systematic review and meta-analysis.

**Figure 2 jpm-13-01308-f002:**
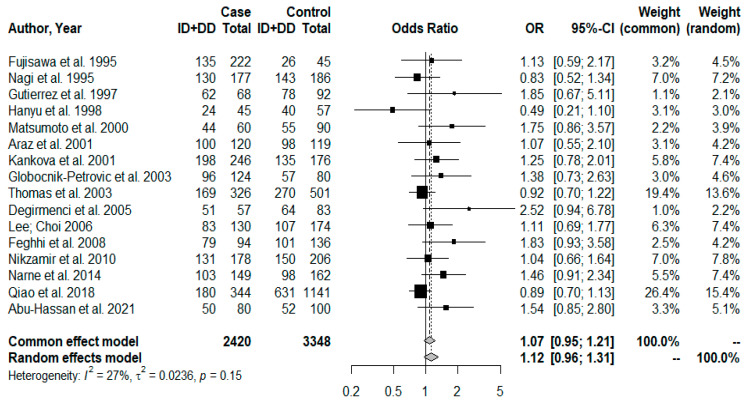
Forest plot for genotypic comparison (II vs. ID+DD). The odds ratio (OR) and 95% confidence interval (95% CI) were calculated using the DerSimonian–Laird method (random effect model) due to the Higgins inconsistency test (I^2^) value. Abu-Hassan et al. 2021 [5], Fujisawa et al. 1995 [16], Nagi et al. 1995 [17], Gutierrez et al. 1997 [18], Hanyu et al. 1998 [19], Matsumoto et al. 2000 [20], Araz et al. 2001 [21], Kankova et al. 2001 [22], Globocnik-Petrovic et al. 2003 [23], Thomas et al. 2003 [24], Degirmenci et al. 2005 [25], Lee; Choi 2006 [26], Feghhi et al. 2008 [27], Nikzamir et al. 2010 [28], Narne et al. 2014 [29], Qiao et al. 2018 [30].

**Figure 3 jpm-13-01308-f003:**
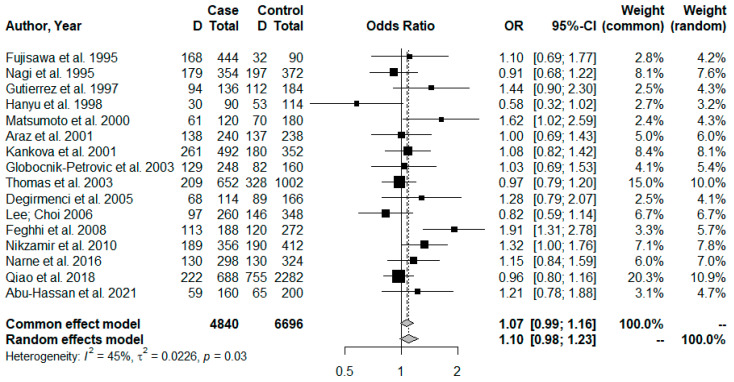
Forest plot for allelic comparison (I vs. D). The odds ratio (OR) and 95% confidence interval (95% CI) were calculated using the DerSimonian–Laird method (random effect model) due to the Higgins inconsistency test (I^2^) value. Abu-Hassan et al. 2021 [5], Fujisawa et al. 1995 [16], Nagi et al. 1995 [17], Gutierrez et al. 1997 [18], Hanyu et al. 1998 [19], Matsumoto et al. 2000 [20], Araz et al. 2001 [21], Kankova et al. 2001 [22], Globocnik-Petrovic et al. 2003 [23], Thomas et al. 2003 [24], Degirmenci et al. 2005 [25], Lee; Choi 2006 [26], Feghhi et al. 2008 [27], Nikzamir et al. 2010 [28], Narne et al. 2014 [29], Qiao et al. 2018 [30].

**Figure 4 jpm-13-01308-f004:**
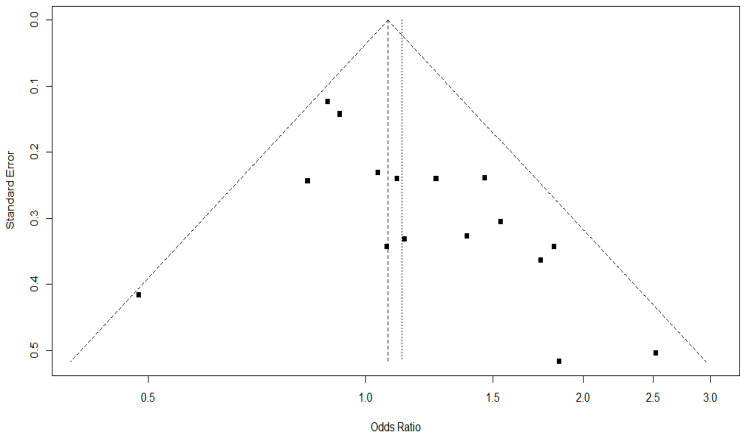
Funnel plot for the publication bias of the studies included in the meta-analysis in the genotypic comparison (II vs. ID+DD). Each square represents a study included in the meta-analysis.

**Figure 5 jpm-13-01308-f005:**
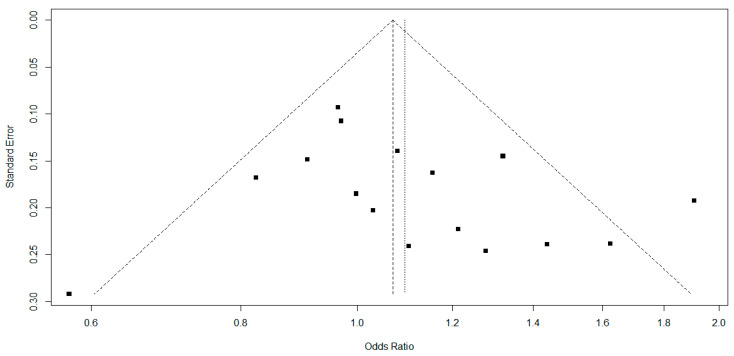
Funnel plot for the publication bias of the studies included in the meta-analysis in the allelic comparison (I vs. D). Each square represents a study included in the meta-analysis.

**Table 1 jpm-13-01308-t001:** Search strategies elaborated for Systematic Review.

Databases	Search Strategy
PubMed/MEDLINE	((“Diabetes Mellitus”[Mesh] OR “Diabetes Mellitus, Type 2”[Mesh]) AND (“Polymorphism, Genetic”[Mesh] OR “Amplified Fragment Length Polymorphism Analysis”[Mesh])) AND (“Diabetic Retinopathy/complications”[Mesh] OR “Diabetic Retinopathy/genetics”[Mesh])
Embase	‘((‘diabetes mellitus’/exp OR ‘non insulin dependent diabetes mellitus’/exp) AND ‘genetic polymorphism’/exp OR ‘amplified fragment length polymorphism’/exp) AND ‘diabetic retinopathy’/exp AND [1992–2022]/py
Web of Science	((((ALL=(diabetes)) OR ALL=(type 2 diabetes mellitus)) AND ALL=(genetic polymorphism)) OR ALL=(amplified fragment length polymorphism analysis)) AND ALL=(diabetic retinopathy)
Scopus	(TITLE-ABS-KEY (diabetes) OR TITLE-ABS-KEY (type 2 diabetes AND mellitus) AND TITLE-ABS-KEY (genetic AND polymorphism) OR TITLE-ABS-KEY (amplified AND fragment AND length AND polymorphism AND analysis) AND TITLE-ABS-KEY (diabetic AND retinopathy)) AND (LIMIT-TO (PUBYEAR, 2022) OR LIMIT-TO (PUBYEAR, 2021) OR LIMIT-TO (PUBYEAR, 2020) OR LIMIT-TO (PUBYEAR, 2019) OR LIMIT-TO (PUBYEAR, 2018) OR LIMIT-TO (PUBYEAR, 2017) OR LIMIT-TO (PUBYEAR, 2016) OR LIMIT-TO (PUBYEAR, 2015) OR LIMIT-TO (PUBYEAR, 2014) OR LIMIT-TO (PUBYEAR, 2013) OR LIMIT-TO (PUBYEAR, 2012) OR LIMIT-TO (PUBYEAR, 2011) OR LIMIT-TO (PUBYEAR, 2010) OR LIMIT-TO (PUBYEAR, 2009) OR LIMIT-TO (PUBYEAR, 2008) OR LIMIT-TO (PUBYEAR, 2007) OR LIMIT-TO (PUBYEAR, 2006) OR LIMIT-TO (PUBYEAR, 2005) OR LIMIT-TO (PUBYEAR, 2004) OR LIMIT-TO (PUBYEAR, 2003) OR LIMIT-TO (PUBYEAR, 2002) OR LIMIT-TO (PUBYEAR, 2001) OR LIMIT-TO (PUBYEAR, 2000) OR LIMIT-TO (PUBYEAR, 1999) OR LIMIT-TO (PUBYEAR, 1998) OR LIMIT-TO (PUBYEAR, 1997) OR LIMIT-TO (PUBYEAR, 1996) OR LIMIT-TO (PUBYEAR, 1995) OR LIMIT-TO (PUBYEAR, 1994) OR LIMIT-TO (PUBYEAR, 1993) OR LIMIT-TO (PUBYEAR, 1992))

**Table 2 jpm-13-01308-t002:** Characteristics of studies included in the systematic review and meta-analysis.

Reference	Country	Design	Case					Control				
			Sample Size	Sex (M/F)	Age (Year)	DM Duration (Year)	Definition	Sample Size	Sex (M/F)	Age (Year)	DM Duration (Year)	Definition
Abu-Hassan et al. [5]	Jordan	Cross-sectional	80	41/41	62 ± 8	10.8 ± 4.1	DR (T2DM)	100	39/61	60 ± 8	7.0 ± 4.3	NDR (T2DM)
Fujisawa et al. [16]	Japan	Case-control	222	NA	NA	NA	DR (T2DM)	45	NA	NA	NA	NDR (T2DM)
Nagi et al. [17]	United Kingdom	Cross-sectional	177	55/45	66.8 ± 10.4	11 (1–36)	DR (T2DM)	186	52/48	69.5 ± 11.1	7 (1–45)	NDR (T2DM)
Gutiérrez et al. [18]	Spain	Case-control	68	29/39	61.90 ± 9.1	14.8 ± 5.7	DR (T2DM)	92	45/47	59.6 ± 10.3	12.1 ± 6.3	NDR (T2DM)
Hanyu et al. [19]	Japan	Case-control	45	15/30	58.0 ± 8.8	18.2 ± 5.7	DR (T2DM)	57	31/26	56.4 ± 5.1	NA	Healthy
Matsumoto et al. [20]	Japan	Case-control	60	27/37	56.8±11.90	16.2 ± 9.1	DR (T2DM)	90	50/40	58.9 ± 12.1	15.0 ± 6.6	NDR (T2DM)
Araz et al. [21]	Turkey	Cross-sectional	120	49/71	55 ± 8	11.2 ± 6.5	DR (T2DM)	119	37/82	51 ± 9	5.2 ± 5.1	NDR (T2DM)
Kanková et al. [22]	Czech Republic	Cohort	246	118/128	64.1 ± 11.1	10.2 ± 9.2	DR (T2DM)	176	67/109	63.60 ± 13.4	NA	Healthy
Globocnik-Petrovic et al. [23]	Slovenia	Case-control	124	59/65	65.6 ± 9.7	18.7 ± 9.1	DR (T2DM)	80	40/40	71.3 ± 7.0	16.8 ± 6.8	NDR (T2DM)
Thomas et al. [24]	China	Case-control	326	128/198	59.8 ± 11.4	6.3 (5.6–7.0)	DR (T2DM)	501	197/304	60.4 ± 9.3	6.0 (5.6–6.3)	NDR (T2DM)
Degirmenci et al. [25]	Turkey	Case-control	57	95/48	57.66 ± 0.85	10.64 ± 0.22	DR (T2DM)	83	83/50	46.39 ± 1.52	NA	NDR (T2DM)
Lee; Choi [26]	South Korea	Cross-sectional	130	77/53	53.1 ± 12.3	11.4 ± 3.7	DR (T2DM)	174	102/72	53.7 ± 12.9	9.4.± 2.8	NDR (T2DM)
Feghhi et al. [27]	Iran	Case-control	94	56/38	60.55 ± 8.15	14.01 ± 4.38	PDR (T2DM)	136	87/49	60.46 ± 7.85	13.40 ± 5.0	NPDR (T2DM)
Nikzamir et al. [28]	Iran	Case-control	178	92/86	59.0 ± 8.7	13 (4–30)	DR (T2DM)	206	114/92	59.5 ± 8.2	11 (1–30)	NDR (T2DM)
Narne et al. [29]	India	Case-control	149	47/102	52.7 ± 7.3	14.7 ± 4.7	DR (T2DM)	162	49/113	53.4 ± 5.4	15.9 ± 5.6	NDR (T2DM)
Qiao et al. [30]	China	Cross-sectional	344	148/197	60.08 ±11.34	8.97 ±7.13	DR (T2DM)	1141	455/691	51.42 ±13.59	4.45 ±4.62	NDR (T2DM)

DM: Diabetes mellitus; DR: diabetic retinopathy; F: female; M: male; NA, not available; NPDR: non-proliferative diabetic retinopathy; NDR: No diabetic retinopathy; PDR: proliferative diabetic retinopathy; and T2DM: type 2 diabetes mellitus.

**Table 3 jpm-13-01308-t003:** Summary of responses for each study included in the risk of bias assessment.

Reference	Q1	Q2	Q3	Q4	Q5	Q6	Q7	Q8	Q9	Q10	Q11
Abu-Hassan et al. [5] ^2^	Y	Y	N	Y	Y	U	Y	Y	NA	NA	NA
Fujisawa et al. [15] ^1^	Y	Y	Y	Y	Y	N	N	Y	Y	Y	NA
Nagi et al. [17] ^2^	Y	Y	Y	Y	Y	U	Y	Y	NA	NA	NA
Gutiérrez et al. [18] ^1^	Y	Y	Y	Y	Y	Y	Y	Y	Y	Y	NA
Hanyu et al. [19] ^1^	Y	Y	Y	Y	Y	Y	Y	Y	Y	Y	NA
Matsumoto et al. [20] ^1^	Y	Y	Y	Y	Y	Y	Y	Y	Y	Y	NA
Araz et al. [21] ^2^	Y	Y	Y	Y	Y	Y	Y	Y	NA	NA	NA
Kanková et al. [22] ^3^	Y	Y	Y	Y	Y	U	Y	Y	Y	NA	Y
Globocnik-Petrovic et al. [23] ^1^	Y	Y	Y	Y	Y	Y	Y	Y	Y	Y	NA
Thomas et al. [24] ^1^	Y	Y	U	Y	Y	Y	N	Y	Y	Y	NA
Degirmenci et al. [25] ^1^	Y	Y	Y	Y	Y	Y	Y	Y	Y	Y	NA
Lee; Choi [26] ^2^	Y	Y	Y	Y	Y	Y	Y	Y	NA	NA	NA
Feghhi et al. [27] ^1^	Y	Y	Y	Y	Y	Y	Y	Y	Y	Y	NA
Nikzamir et al. [28] ^1^	Y	Y	Y	Y	Y	Y	Y	Y	Y	Y	NA
Narne et al. [29] ^1^	Y	Y	Y	Y	Y	Y	Y	Y	Y	Y	NA
Qiao et al. [30] ^2^	Y	Y	Y	Y	Y	Y	Y	Y	NA	NA	NA

^1^ Case-control; ^2^ Cross-sectional; ^3^ Cohort. N—No; NA—Not applicable; U—Unclear; Q—Question; and Y—Yes.

**Table 4 jpm-13-01308-t004:** Genotypic and allelic frequencies of the *ACE* gene (I/D) polymorphism were included in the systematic review and meta-analysis.

Reference	Genotype Frequency						Allele Frequency			
	I/I		I/D		D/D		I		D	
	Case	Control	Case	Control	Case	Control	Case	Control	Case	Control
Abu-Hassan et al. [5]	30	48	41	39	9	13	101	135	59	65
Fujisawa et al. [16]	87	19	102	20	33	6	276	58	168	32
Nagi et al. [17]	47	43	81	89	49	54	175	175	179	197
Gutiérrez et al. [18]	6	14	30	44	32	34	42	72	94	112
Hanyu et al. [19]	21	17	18	27	6	13	60	61	30	53
Matsumoto et al. [20]	16	35	27	40	17	15	59	110	61	70
Araz et al. [21]	20	21	62	59	38	39	102	101	138	137
Kanková et al. [22]	48	41	135	90	63	45	231	172	261	180
Globocnik-Petrovic et al. [23]	28	23	63	32	33	25	119	78	129	82
Thomas et al. [24]	157	231	129	212	40	58	443	674	209	328
Degirmenci et al. [25]	6	19	34	39	17	25	46	73	68	89
Lee; Choi [26]	47	67	69	68	14	39	163	202	97	146
Feghhi et al. [27]	15	35	45	82	34	19	75	152	113	120
Nikzamir et al. [28]	47	56	73	110	58	40	167	222	189	190
Narne et al. [29]	46	64	76	66	27	32	168	194	130	130
Qiao et al. [30]	164	510	138	507	42	124	466	1527	222	755
Total	785	1243	1123	1524	512	581	2693	4006	2147	2686

D: Deletion; and I: Insertion.

**Table 5 jpm-13-01308-t005:** Publication bias test results for genotypic and allelic analyses.

Analyzes Performed	Egger’s Test	Begg’s Test
Genotypic	0.01	0.05
Allelic	0.25	0.27

## Data Availability

No new data were created or analyzed in this study. Data sharing does not apply to this article.

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
