# Peer review of "No Association of Angiotensin-Converting Enzyme Insertion/Deletion (ACE I/D) Gene Polymorphism in the Susceptibility to Diabetic Retinopathy in Type 2 Diabetes Mellitus Patients: An Updated Meta-Analysis"

_jpm, 2023, doi:10.3390/jpm13091308_

Round 1

Reviewer 1 Report

This manuscript (jpm-2523914) presents a comprehensive meta-analysis investigating the potential association between the genetic polymorphism rs1799752 of angiotensin-converting enzyme and diabetic retinopathy.  The literature has reported inconsistent conclusions, and to address this, the authors selected sixteen studies based on specific criteria outlined in Table 1.  Genotypic and allelic frequencies were combined for meta-analyses (Table 4).  The meta-analyses of these previous publications (Fig. 2 to 5) indicate no significant association between this polymorphism and the risk of developing diabetic retinopathy.

However, the current version's presentation of results does not meet the necessary quality for publication.

Major Concerns:

1. Title: The paper's title should clearly reflect the research's conclusion, indicating that no association exists between the genetic polymorphism rs1799752 and diabetic retinopathy.

2. Results Section: The authors should expand the content of the Results section, particularly regarding the meta-analysis (Section 3.2).  Additional insights could be provided on how the sample size of a study affects the documentation of Odds Ratio in Fig. 2 and Fig. 3.  Describing whether Funnel plots have reasonable dot distribution patterns in Fig. 4 and 5 would be beneficial.  These descriptions would help readers understand why discrepancies occur among studies and identify which types of studies may offer more credible conclusions, as discussed in the second to fourth paragraphs of Page 12.

3. Abstract: The abstract should be revised to clearly present the findings of the meta-analyses.

Minor Issue:

Lines 197 to 199 at Page 10 appear to be sentences that were inadvertently left in the manuscript and should be removed.

Author Response

ID JPM-2523914 - Genetic polymorphism of angiotensin-converting enzyme and diabetic retinopathy risk in type 2 diabetes mellitus patients: A Systematic Review and Meta-analysis.

Answers to the reviewer requirements:

REVIEWER 1:

1

Title: The paper's title should clearly reflect the research's conclusion, indicating that no association exists between the genetic polymorphism rs1799752 and diabetic retinopathy.

Done. We changed and following the reviewer suggestion. It was added in the title “an update” to improve the visibility this meta-analysis.

2

The authors should expand the content of the Results section, particularly regarding the meta-analysis (Section 3.2). Additional insights could be provided on how the sample size of a study affects the documentation of Odds Ratio in Fig. 2 and Fig. 3.  Describing whether Funnel plots have reasonable dot distribution patterns in Fig. 4 and 5 would be beneficial. These descriptions would help readers understand why discrepancies occur among studies and identify which types of studies may offer more credible conclusions, as discussed in the second to fourth paragraphs of Page 12.

Done.

Described in lines 176 to 186.

 We observe that the larger the sample size, the greater the weight of the study and the smaller the CI, indicating greater certainty in the results with a larger sample size (Figures 2 and 3). Small samples may reflect observations due to chance, so the sta-tistical indicators used in meta-analyses, such as CI and individual weight, confirm the degree of reliability of the results.

Our findings showed moderate heterogeneity in the Higgins inconsistency test, which did not interfere with the results obtained. However, we can observe the large variation in sample sizes of the included studies (Table 2), which may reflect the degrees of divergence found. In addition, the funnel plot estimated the publication bias for genotypic comparison (Figure 4) and allelic comparison (Figure 5), and no significant asymmetries were found to identify publication bias

3

The abstract should be revised to clearly present the findings of the meta-analyses.

Done.

Lines 197 to 199 at Page 10 appear to be sentences that were inadvertently left in the manuscript and should be removed.

Done

Reviewer 2 Report

1. In the meta-analysis there are mentioned 30 publications between 1992-2022. Could you please sum up how many publications per year? I would be interested to know how recent they are most of them.

2. Did the authors find any common genetic characteristics related to the ethnicity of the subjects? There is a brief mention i lines 212 and 213, but could you maybe, detail?

3. Conclusion should be extended more, maybe add lines 288 to 298 to the Conclusions paragraph

4. Overall, a very interesting and complexe meta-analysis

satisfying quality of English

Author Response

ID JPM-2523914 - Genetic polymorphism of angiotensin-converting enzyme and diabetic retinopathy risk in type 2 diabetes mellitus patients: A Systematic Review and Meta-analysis.

Answers to the reviewer requirements:

REVIEWER 2:

1

In the meta-analysis there are mentioned 30 publications between 1992-2022. Could you please sum up how many publications per year? I would be interested to know how recent they are most of them.

Done. We added and following the reviewer suggestion (lines 142 to 144).

We identified two studies published in the years 1995, 2001 and 2003, respectively, and one article was published in each of the years 1997, 1998, 2000, 2005, 2006, 2008, 2010, 2014, 2018 and 2021

2

Did the authors find any common genetic characteristics related to the ethnicity of the subjects? There is a brief mention i lines 212 and 213, but could you maybe, detail?

Done.

Described in lines 224 to 226.

 In this systematic review, we only observed European and Asian studies, making it impossible to characterize and analyze specific genetic factors for each ethnicity, thus revealing the lack of studies analyzing the relationship between ACE gene (I/D) polymorphisms and the DR risk in different populations.

3

Conclusion should be extended more, maybe add lines 288 to 298 to the Conclusions paragraph

Done.

4

Overall, a very interesting and complexe meta-analysis

We would like to thank you to incorporate the suggestions to improve the manuscript.

Round 2

Reviewer 1 Report

This revision (jpm-2523914) presents a meta-analysis aimed at resolving the discrepancy among studies concerning the potential association between the genetic polymorphism rs1799752 of angiotensin-converting enzyme and diabetic retinopathy. The changes suggested by this reviewer have been incorporated into this version. However, prior to publication, a few minor issues need to be addressed.

Firstly, it is proposed to add the following sentence to the abstract: "Forest plot analysis revealed that the discrepancy between previous studies most likely arises from variations in their sample sizes."

Secondly, there are inconsistencies in the data presented in Table 4 compared to other tables and figures. This discrepancy appears to result from the inclusion of one additional item, i.e., [Gutiérrez et al.], in the first column.

Author Response

In this revision (jpm-2523914) all comments and suggestions were done.
Thank you
Best regards
Angela Adamski